# Factors Involved in Endothelial Dysfunction Related to Angiogenic Disbalance and Oxidative Stress, in Women at High Risk of Term Pre-Eclampsia

**DOI:** 10.3390/antiox11071409

**Published:** 2022-07-21

**Authors:** Jean Michell Santoyo, José Antonio Noguera, Francisco Avilés, Juan Luis Delgado, Catalina de Paco-Matallana, Virginia Pérez, Isabel Hernández

**Affiliations:** 1Physiology Department, Institute of Biomedical Research (IMIB-Arrixaca), Universidad de Murcia, 30120 Murcia, Spain; jm.santoyogarcia@um.es; 2Institute of Biomedical Research (IMIB-Arrixaca), Hospital Clínico Universitario Virgen de la Arrixaca, 30120 Murcia, Spain; ja.noguera@gmail.com (J.A.N.); fvaviles@yahoo.es (F.A.); juanluisdelgado@um.es (J.L.D.); katydepaco@hotmail.com (C.d.P.-M.); 3Departamento de Ciencias Sociosanitarias, Institute of Biomedical Research (IMIB-Arrixaca), Universidad de Murcia, 30120 Murcia, Spain; virperez@um.es

**Keywords:** oxidative stress, inflammation, pre-eclampsia, endothelial function prevention women’s health

## Abstract

Oxidative and inflammatory stress, angiogenic imbalance, and endothelial dysfunction are pathophysiological mechanisms occurring in pre-eclampsia (PE) that may persist over time and predispose women to a higher risk of cardiovascular disease (CVD) in the future. However, there is little evidence on the vascular function of women at risk of PE who have not developed the disease. The main objective of this research is to study factors and biomarkers involved in endothelial dysfunction related to oxidative stress, angiogenic disbalance, and inflammation in women at high risk of term PE who do not develop the disease. An observational, analytical, retrospective, and descriptive study was carried out in a selected sample of 68 high-risk and 57 non-risk of term PE participants in the STATIN study (FFIS/2016/02/ST EUDRACT No: 2016-005206-19). A significant increase in mean arterial pressure (MAP) levels and oxidative stress biomarkers (uric acid, homocysteine, and total serum antioxidant capacity) was found. Biomarkers of inflammation (interleukin-6 and growth differentiation factor 15) and endothelial function (asymmetric dimethylarginine) were significantly elevated in the group at risk of pre-eclampsia. A significative dependence relationship was also established between MAP and interleukin-6 and uric acid. These results suggest that women at high risk of term PE may represent pregnancies with pre-existing maternal risk factors for CVD, manifested by the own cardiovascular overload of pregnancy. A better understanding of maternal cardiovascular function in pregnancy would allow the improved prediction of CVD late in life in women.

## 1. Introduction

Pre-eclampsia (PE) is a multifactorial and multisystemic disease that complicates 2–8% of pregnancies and constitutes an important cause of maternal and neonatal morbidity and mortality. Its pathophysiology is associated with generalized maternal endothelial dysfunction due to the interaction of placental and maternal factors.

The two-stage placental model of PE suggests that both early- and late-onset PE result from placental syncytiotrophoblast stress. This stress emerges from the convergence of placental dysfunction Stage 1 processes, promoting the clinical stage 2 of PE (new-onset hypertension and proteinuria or other signs of end-organ dysfunction), but the causes and timing of placental malperfusion differ. This model includes most risk factors for PE, such as primiparity, chronic disease prior to pregnancy (e.g., obesity, diabetes, chronic hypertension, and autoimmune diseases), and factors of pregnancy risk (e.g., multiple or molar pregnancies and gestational diabetes), or hypertension and low circulating placental growth factor (PlGF). These factors may increase the risk of progressing to second-stage PE by affecting pathways leading to Stage 1, as well as potentially accelerating steps towards Stage 2, in addition to increasing maternal cardiovascular susceptibility to inflammatory factors released by the placenta. The two-stage theory of the pathophysiology of PE proposes that an increased release of soluble fms-like tyrosine kinase-1 (sFlt-1) antiangiogenic factor into the maternal circulation is responsible, at least in part, for the vascular endothelial dysfunction characteristic of the second stage [1].

Different studies show the presence of systemic vascular dysfunction in women with PE that persists over time [2,3,4]. Induced flow-mediated vasodilation is impaired in pregnant women before and during developing PE, as well as 3 years after delivery, indicating vascular endothelial dysfunction, which is present in the pathophysiology of PE and remains after the disappearance of the clinical symptoms, indicating a possible association with an increased risk of future cardiovascular disease (CVD) [5]. One of the limitations of many of the existing studies on vascular dysfunction after PE is the lack of knowledge about vascular function in women before they are affected by PE. This raises the question of whether pregnancy in these women is unmasking a previous predisposition to developing CVD. For this reason, we propose to study the existence of factors involved in endothelial dysfunction related to angiogenic disbalance and oxidative stress in women diagnosed with a high risk of term PE who do not develop the disease. The aim of this work is to assess the alteration in biomarkers of oxidative stress (uric acid, homocysteine and total antioxidant capacity), inflammation (Interleukin-6, high-sensitivity C-reactive protein, and Growth differentiation factor 15), and endothelial function (asymmetric dimethylarginine) in the plasma of pregnant women, diagnosed as at high risk of term PE.

## 2. Methods

Eligible participants were randomly selected through a term PE screening study conducted as part of a double-blind, controlled clinical trial (STATIN, EC Nº EducraCT 2016-005206-19, ISRCTN16123934, https://doi.org/10.1186/ISRCTN16123934), among women attending their routine hospital visit at Virgen de la Arrixaca University Clinical Hospital (HCUVA) with 35 + 0—36 + 6 gestation’s weeks. Maternal characteristics and medical and obstetric history were recorded. MAP was determined by validated devices and standardized protocols. The uterine arteries were visualized by transabdominal color doppler ultrasonography, where the pulsatility index of the right and left arteries was measured, and their mean value and their expression in MoM were calculated.

The effective screening for term PE was performed based on the Bayesian model with a detection rate of 75% and a screen-positive rate of 10% [6,7]. This model includes a combination of maternal factors with measures of MAP and a determination of serum PlGF and sFlt-1. Pregnancies identified as being at high risk of term PE (≥1/20) were part of the high-risk group; those considered as being at low risk (<1/20) were included in the control group.

Inclusion criteria for the study were age ≥ 18 years, singleton pregnancy, and live fetus. Exclusion criteria were the existence of major fetal abnormality, PE established, and congenital abnormalities. Potential trial participants were given written information, and those who agreed to participate provided written informed consent. The study was conducted according to the guidelines of the Declaration of Helsinki and approved by the Drug Research Ethics Committees of Virgen de la Arrixaca University Clinical Hospital (HCUVA), 29 November 2018, and Universidad de Murcia.

Blood was drawn from an antecubital arm vein after fasting overnight. Serum and EDTA or citrate plasma were collected and stored at −80 °C at Region de Murcia’s Biobank (PT17/0015/0038).

The lipid profile, uric acid (UA), and high-sensitivity C-reactive protein (hs-CRP) were determined using automated methods on a cobas c8000 modular platform (Roche Diagnostics^®^ International Ltd., Rotkreuz, Switzerland). LDL cholesterol was calculated using Friedewald’s formula. Apo B, Apo A-I, Lp A, and HCy concentrations were determined by immunonephelometry on a BN ProSpec compact autoanalyzer (Siemens Healthcare Diagnostics^®^, Margurg, Germany). A fully automated B·R·A·H·M·S KRYPTOR Compact Plus system (Thermo Fisher Scientific, Waltham, MA, USA) was used to perform sFlt-1 and PlGF assays, both being homogeneous sandwich immunoassays based on the Time-Resolved Amplified Cryptate Emission (TRACE) technology. To assay antioxidant capacity in plasma, we used a Total Antioxidant Status Assay Kit (Cat. 615700, de Sigma-Aldrich, St. Louis, MI, USA) on an ILab 650 Chemistry System (Diamond Diagnostics Inc., Holliston, MA, USA). Interleukin-6 (IL-6) and Growth differentiation factor 15 (GDF-15) levels were determined by electrochemiluminescence immunoassays using a cobas e411 analyzer (Roche Diagnostic^®^). A competitive-inhibition enzyme-linked immunosorbent assay (cELISA) was used to measure Asymmetric dimethylarginine (ADMA) concentrations (Cat. No. KSB301Ge11, Cloud-Clone Corporation, Houston, TX, USA).

Data were reported as mean and standard deviations for normally distributed numerical data or median with interquartile ranges for not normally distributed ones; categorical data were expressed as frequencies with percentages. The association between the different variables studied was determined using Spearman’s correlation coefficient. Differences were analyzed by unpaired Student’s *t*-test or Mann–Whitney U test when appropriately significant. Linear regression analyses were used to study how the change in sFlt-1 affects the studied variables of oxidative stress, inflammation, and endothelial dysfunction, and to see how changes in the variables of oxidative stress, inflammation, and endothelial dysfunction affect MAP. A *p* < 0.05 was considered statistically significant. Statistical analyses were performed using the Stata v15 program, College Station, TX, USA.

## 3. Results

### 3.1. Demographic and Clinical Characteristics of the Participants

Characteristics of the study population are summarized in Table 1. A total of 125 pregnant women were recruited between August 2018 and November 2019, with 35–36 weeks of gestational age. In total, 57 uncomplicated pregnant women were included as a control group (low-risk no PE), and 68 women were included in the high-risk group of term PE. Of the latter, 61 pregnant at high risk did not develop PE (high-risk no PE), and 7 pregnant women did (High-risk PE).

The mean age of the participants was similar to the fertility indicators of the National Institute of Statistics (INE; SPAIN) published in 2019. On the other hand, 42.1% of the pregnant women in the control group were over 35 years old, while 44.1% in the high-risk group met this condition. Gestational age was higher in the high-risk PE group. There were no statistically significant differences in body weight, although body mass index was significantly higher in women with high risk of term PE. Additionally, 44% of women in the high-risk group had a BMI > 30 kg/m^2^, compared to 22% in the control group. As expected, a higher proportion of pregnant women in the high-risk group were primiparous (69% vs. 50%; *p* < 0.05) and had a familiar history of PE (*p* = 0.031). None of the selected women consumed alcohol or drugs. In addition, 18% of the high-risk pregnancies developed gestational diabetes, compared to 1.8% of women in the control group (*p* < 0.001). In our study we found no significant differences between groups in the Uterine artery pulsatility index (UtA-PI). These data agree with the results of Panaitescu, who reported that the inclusion of UtA-PI in the Bayesian model does not improve its efficacy in predicting term PE [8]. High-risk women had significantly higher blood pressure compared to low-risk women (*p* < 0.001). When we separated the high-risk pregnant women into those who had PE and those who did not, we found that those who did develop PE had a significantly higher level of PAM 101 (96–107) at the start of the study than those who did not 94 (91–98) mmHg. However, at this time, none of the systolic/diastolic values were above 140/90 mmHg.

Regarding pregnancy outcomes, it can be highlighted that women at high risk had spontaneous labors in lower numbers and showed a significatively lower proportion of vaginal deliveries than those at low risk. Seven of the high-risk group developed PE diagnosed at the time of delivery (all having systolic/diastolic blood pressures > 140/90: six of them with +2 in the urine protein dipstick test, and one with elevated liver enzymes). Seven other women had pregnancy-induced hypertension.

Laboratory data of the groups with low and high risk of PE, as well as those of pregnant women at high risk diagnosed with PE, are shown in Table 2. There were no significant differences in the lipid profiles between the groups with low and high risk of PE. However, we found significantly decreased values, as well as an increased ApoB/A1 ratio, in pregnancies that developed PE in respect to those in the high-risk no PE group. As expected, the levels of PlGF and sFlt-1 were significantly lower and higher, respectively, in the high-risk no PE group compared to the control group. In addition, significantly higher levels of proteinuria were detected in patients with PE, and a certain increase in serum creatinine values could be seen, although statistical significance was not reached.

### 3.2. Biomarkers Related to Vascular Dysfunction Associated with PE and CVD

In this study, the measurement of oxidative stress markers showed significant differences between groups (Figure 1). As can be seen in Figure 1A, plasma UA levels were significantly higher in the high-risk group that did not develop PE (high risk no PE) than in the low-risk PE group, 4.6 (1.1–5.1) mg/dL vs. 3.7 (3.2–1.1) mg/dL respectively (*p* < 0.001). Figure 1B shows significantly higher levels of Hcy 6.3 (5.7–7.1) µmol/L in pregnant women of the high-risk no PE group compared with those in the low-risk group, 5.3 (4.7–5.9) µmol/L (*p* < 0.001). In consonance with the above results, Figure 1C shows a lower plasma antioxidant capacity in pregnant women in the high-risk no PE group 1.6 (1.5–1.7) mmol/L compared to pregnant women at low PE risk 1.7 (1.6–1.8) mmol/L (*p* < 0.001). However, plasma levels of Hcy 6.5 (4.5–8.0) µmol/L; UA 5.0 (1.5–5.8) mg/dL and total antioxidant capacity (TAC) 1.6 (1.5–1.7) mmol/L in the high-risk PE group were similar to those of the high-risk no PE group.

The results of the measurement of inflammatory and endothelial dysfunction biomarkers in the three studied groups are shown in Figure 2. Pregnant women of the high-risk no PE group have significantly higher levels of ADMA 786 (502–1497) ng/mL than those of the low-risk PE group 656 (537–991) ng/mL (*p* < 0.05). In pregnancies with PE, higher levels of ADMA were observed, 1269 (641–1861) ng/mL, without reaching statistical significance, probably due to the small number of patients (*n* = 7). In this work, while C-reactive protein did not experience changes between groups, we found an elevation of plasma IL-6 and GDF-15 even in patients at high risk who did not develop PE (4.1 (2.8–5.1) pg/mL; 117 (74–145) µg/mL, respectively) versus those at low risk of PE (1.5 (1.5–2.36); 79 (64–117) µg/mL, respectively) (*p* < 0.001).

## 4. Discussion

Women with a history of PE have a higher risk of future cardiovascular, cerebrovascular, and peripheral arterial disease, and cardiovascular mortality [9]. A large proportion of women who experienced PE had major cardiovascular risk factors in the fifth decade of life, compared with healthy controls [10]. Different systematic reviews have revealed the long-term consequences of PE and have quantified the risk of CVD after pregnancy in mothers with PE and their children [11,12]. Pre-eclampsia and high cardiovascular risk share several pathophysiological mechanisms, such as oxidative and inflammatory stress, angiogenic imbalance, and endothelial dysfunction. In animal models and humans, an increase in proinflammatory cytokines and acute-phase reactants has been described after early- and late-onset PE [13]. A recent systematic review shows evidence that sFlt-1, PlGF, IL-6, IL-6/IL-10 ratios are interesting candidate biomarkers for cardiovascular risk stratification after PE or HELLP syndrome (hemolysis, elevated liver enzymes and low platelets) [14]. On the other hand, many pregnant women who are classified as high risk for PE based on obstetric and maternal factors, however, ultimately do not develop the disease. This raises the possibility of the existence of a predisposition to vascular dysfunction that is evident in pregnancies with a high risk of PE, which may also be a risk factor for CVD later in life, regardless of its association to the development of PE. In this study we analyzed different markers of oxidative stress, inflammation, and endothelial dysfunction involved in the pathophysiology of PE and CVD disease in women at high risk of PE, finding a significant elevation of these markers in comparison with low-risk pregnancies.

Over the last decade, substantial progress has been made in understanding the role of oxidative and nitrosative stress in the pathophysiology of PE [15,16]. In this sense, it has been suggested that PE is caused by a placental abnormality, which leads to a decrease in placental perfusion and the occurrence of repeated ischemia/reperfusion episodes, which create a favorable environment for the development of oxidative stress. In turn, this oxidative damage to the placenta produces inflammation, apoptosis, and the release of cellular debris into the maternal circulation, together with various anti-angiogenic factors such as sFlt-1, cytokines, and oxidants, which act on the maternal vascular endothelium, inducing stress. Oxidative stress stimulates the production and secretion of proinflammatory cytokines, as well as vasoactive compounds, resulting in systemic endothelial dysfunction characterized by inflammation and vascular constriction. In fact, oxidative stress appears to be the central component of placental and endothelial dysfunction, the causative etiology of PE [17]. Some studies suggest that high UA levels lead to PE, while others claim that PE causes increased UA levels [18]. In the present study, UA and Hcy levels correlate with MAP. This result is consistent with other reports indicating that UA levels correlate with blood pressure [19] and that hyperuricemia is associated with greater severity of PE [20] and adverse maternal and particularly fetal outcomes [21]. An elevated serum UA concentration positively correlated with Tumor necrosis factor alpha (TNF-α) and soluble Intercellular adhesion molecule-1 (ICAM-1) in pregnant women, and a significant association was found with the inflammation of the maternal systemic vasculature as indicated by increased TNF-α and ICAM-1 expression in the subcutaneous fat tissue of women with PE. Several studies show that many pro-inflammatory factors, including IL-1β, IL-6, IL-7, IL-8, IL-17a, Monocyte Chemoattractant Protein-1, and Macrophage inflammatory protein-1β, were significantly increased in the serum of women with PE compared to controls [22,23,24]. In this sense, when comparing with normal pregnancies, we found an increase in IL-6 levels in women with a high risk of PE regardless of whether or not they developed the disease.

The association of hyperhomocysteinemia and PE was initially suggested by Decker et al. [25,26]. A large number of studies have demonstrated a positive correlation between Hcy and PE and have shown a significant increase in serum Hcy levels in women who develop PE at various stages of pregnancy [27,28,29]. In addition, pregnant women complicated by severe PE have shown significantly higher serum Hcy levels than those in the non-severe form. These data agree with the significantly higher values of maternal serum Hcy found in pre-eclamptic women, compared to those registered in the blood of control mothers, previously reported by our group [30]. Previous works have proposed that hyperhomocysteinemia may contribute to the development of PE by ADMA accumulation, leading to endothelial dysfunction [31,32]. In this sense, our results show an increase in the plasma levels of Hcy and ADMA in the high-risk group without PE. In addition, in the high-risk group that developed PE, an upward trend in the plasma levels of these two biomarkers is observed. ADMA, an analog of L-arginine, is an endogenous metabolite that competitively inhibits L-arginine for all three isoforms of Nitric oxide synthase (NOS). Elevated levels of ADMA block Nitric oxide synthesis and decrease cellular uptake of L-arginine, which is recognized as a biomarker of endothelial dysfunction. ADMA levels are lower in normal pregnant women than in non-pregnant women. In pregnancies complicated by PE, ADMA levels are significantly higher than in the control group of the same gestational age [33,34,35], and a higher concentration of ADMA in patients with early-onset PE is related to the severity of PE [36]. ADMA has been shown to increase systemic vascular resistance in humans by acting as an endogenous inhibitor of NOS. An elevation of ADMA levels has been widely described in patients with various cardiovascular and metabolic conditions, such as hypercholesterolemia, atherosclerosis, hypertension, chronic heart or kidney failure, diabetes mellitus, and stroke [27].

As noted above, an association between MAP values and laboratory parameters was found. The dependence relationships of MAP with IL-6, UA and Hcy values, in the third trimester of pregnancy, are shown in Table 3. Both R2 and the adjusted R2 explain that about 16% of the variation in MAP is due to IL-6, UA, and Hcy levels. In the regression, the corresponding coefficients, 2.06 for UA and 1.114 for IL-6, represent the predicted change in MAP for every unit change in each of these variables, when the other variable remains constant.

In recent years, evidence has accumulated tha suffer from PE predisposes to increased cardiovascular risk long-term in life [36], including risk of hypertension, peripheral arterial disease, coronary artery disease, and cerebrovascular disease [37]. The relationship between CVD and PE lies in endothelial dysfunction, with many studies suggesting that the persistence of this dysfunction postpartum is what contributes to the onset of diseases such as diabetes, hypertension, and increased risk of CVD [4]. PE and CVD share risk factors such as obesity, premature familial CVD, arterial hypertension, oxidative stress, and inflammation, having as a common basis the presence of this endothelial dysfunction [38]. In addition, further investigations focused on the prospective assessment of inflammation and endothelial dysfunction biomarkers in women with PE have been proposed to find out if these markers are capable of distinguishing women who subsequently develop hypertension and CVD or not. From the summary of evidence, sFLT-1, PlGF, IL-6, IL-6/IL-10 ratio, high-sensitivity cardiac troponin I, activin A, soluble human HLA-G, PAPP-A, and norepinephrine show potential and are interesting candidate biomarkers to further explore [14].

In the present study, pregnant women diagnosed as being at high risk of PE, who did not develop it, had similar levels of the studied biomarkers to those who did develop PE. This finding suggests the existence of a vascular dysfunction in non-developing PE high-risk pregnant women that may also predispose them to a higher cardiovascular risk throughout life. Furthermore, women at high risk of PE had higher levels of MAP than those at low risk. Given these results, elevated oxidative stress and inflammation biomarkers in pregnant women with a high risk of PE, regardless of whether they develop it or not, makes it possible to detect women with high cardiovascular risk, to predict the risk of future cardiovascular events, and provide a useful tool for monitoring therapeutic efficacy.

Taking all the above results together, it could be suggested that women at high risk of term PE may represent pregnancies with pre-existing maternal risk factors that reach Stage 1, placental dysfunction stress, but not Stage 2, of PE development. The question remains if there is an opportunity for intervention in asymptomatic women with a history of high risk of PE. The identification of circulating cardiovascular biomarkers of relevance for myocardial and coronary artery function such as UA, Hcy, IL-6 and ADMA may, therefore, be of additional value to determine which women are at greatest risk, and to better understand the pathophysiological mechanism of CVD later in life in young women. Then, this opens an opportunity to establish measures to prevent CVD in this population.

## Figures and Tables

**Figure 1 antioxidants-11-01409-f001:**
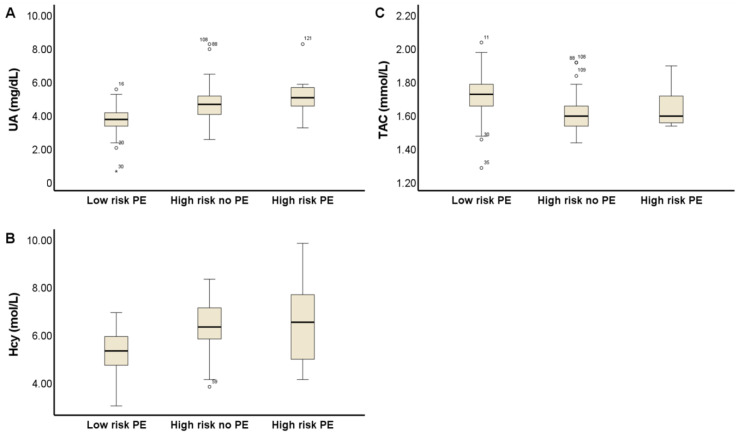
Box and whisker plots showing plasma levels of oxidative stress biomarkers in the study groups. AU, Uric acid (**A**); Hcy, Homocysteine (**B**); and TAC, Total antioxidant capacity (**C**).

**Figure 2 antioxidants-11-01409-f002:**
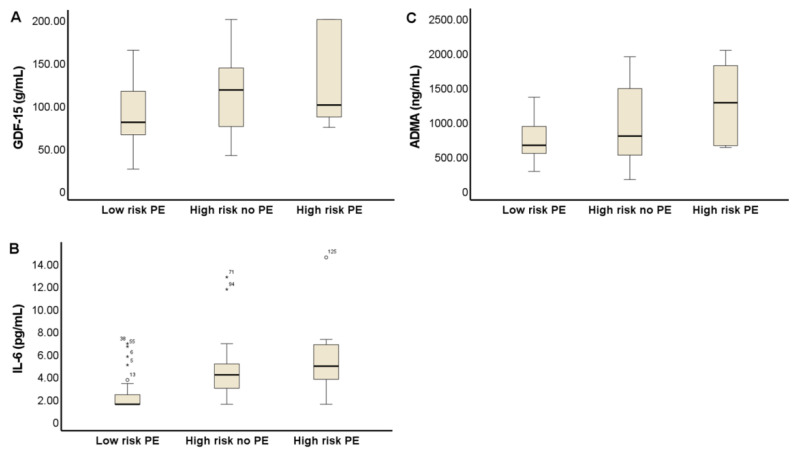
Box and whisker plots of plasma levels of inflammation and endothelial dysfunction biomarkers in the study groups. GDF-15, Growth differentiation factor 15 (**A**); IL-6, Interleukine-6 (**B**); and ADMA, asymmetric dimethylarginine (**C**).

**Table 1 antioxidants-11-01409-t001:** Characteristics of the Participants.

Characteristic	Low Risk	High Risk	*p* Value
	(*n* = 57)	(*n* = 68)	
Age, years	32.6 ± 5.7	32.8 ± 6.4	0.76
Weight, kg	73.5 (67.5–79.2)	76.25 (70–84.9)	0.152
Height, cm	163 (160–166.5)	160 (157–164.5)	0.044
Body mass index, kg/m^2^	28.1 ± 3.2	30.1 ± 4.7	0.016
Caucasian	57 (100)	66 (97.1)	0.294
Cigarette smoker	10 (17.5)	5 (7.4)	0.071
Nulliparous,	29 (50.9)	47 (69.1)	0.029
Family history of PE	1 (1.8)	8 (11.8)	0.031
*Medical History*			
Lupus	1 (1.8)	0	0.456
AAS	1 (1.8)	1 (1.5)	0.706
Diabetes mellitus	0	1 (1.5)	0.544
Hypertension	0	2 (2.9)	0.294
*Obstetric History*			
In vitro fertilization	6 (10.5)	12 (17.6)	0.192
Gestational age, weeks	35.1 (35–35.4)	35.4 (35.3–35.7)	
MAP, mm Hg	86.3 (79.1–89.9)	95.5 (91.0–100.2)	*p* < 0.0001
UtA-PI	0.68 (0.6–0.81)	0.69 (0.6–0.86)	0.392
UtA-PI (MoM)	0.95 (0.83–1.13)	0.99 (0.85–1.16)	0.319
Gestational diabetes	1 (1.8)	8 (11.8)	0.031
Risk of PE	292 (102–876)	9 (6–15)	*p* < 0.0001
*Pregnancy Outcomes*			
Labor			
Spontaneous	40 (70.2)	30 (44.1)	0.004
Induced	12 (21.1)	34 (50)	0.001
No labor	5 (8.7)	4 (5.9)	0.535
Mode of delivery			
Vaginal	37 (64.9)	30 (44.1)	0.021
Cesarean	12 (21.1)	21 (30.9)	0.216
Instrumental	8 (14)	17 (25)	0.128
GA at delivery (w)	39.9 (39.3–40.6)	39.8 (39.1–40.4)	0.517
Birth weight (g)	3360 (3052–3577)	3218 (2929–3537)	0.311
PE	0	7 (10.3)	
PI	0	7 (10.3)	

Data are reported as mean ± standard deviation, median (interquartile range) or number (percentage). AAS, Antiphospholipid antibody syndrome; Mean arterial pressure, MAP; Uterine artery pulsatility index, UtA-PI. Gestational age; PIH, pregnancy induced hypertension.

**Table 2 antioxidants-11-01409-t002:** Laboratory data in pregnant women at low risk, high risk, and term PE.

Laboratory Parameters	Low Risk PE	High Risk No PE	High Risk PE
	(*n* = 57)	(*n* = 61)	(*n* = 7)
*Dyslipidemia*	median (IQR)	median (IQR)	median (IQR)
Cholesterol, (mg/dL)	248.7 (217.5–280.5)	245 (205–276)	253 (211–266)
Triglyicerides (mg/dL)	246 (189.5–292)	258 (226–303)	248 (137–263)
HDL (mg/dL)	76 (63.5–82.5)	68 (58–78)	59 (45–74)
LDL (mg/dL)	127 (102–151)	121 (90.5–147.25)	124 (119–159)
Apo B (mg/dL)	133 (115–153)	126 (110–155)	137 (118–161)
Apo A-I (mg/dL)	224 (205–246.5)	218 (203–232)	200 (158–226) ꭊ
ApoB/Apo A-I ratio	0.61 (0.52–0.69)	0.58 (0.50–0.71)	0.78 (0.58–0.89) ꭊ
Lp A (mg/dL)	13.5 (4.7–27.45)	13 (7.6–32.2)	17.1 (4.8–31.3)
*Angiogenic*			
PlGF (pg/mL)	354.7 (165.5–546.3)	88.6 (56.3–130.7) *	122.6 (77.74–489.9)
PlGF MoM	1.05 (0.60–1.73)	0.32 (0.23–0.56) *	0.45 (0.33–1.11)
sFlt-1 (pg/mL)	2090 (1489–2689)	5027 (3723–6452) *	4902 (1972–7259)
sFlt-1 (MoM)	1.03 (0.71–1.36)	2.10 (1.72–2.84) *	2.06 (1.02–5.33)
sFlt-1/PLGF	5.6 (3.6–14)	58.1 (35.8–87.3) *	23.5 (3.9–134.3)
*Other Parameters*			
Proteinuria (mg/dL)	10 (7.0–11.7)	10 (8–14)	93.5 (17.5–225) ꭊ
Platelets (×1000/mm^3^)	217 (180–217)	199 (163–249)	249 (186–279)
Creatinin mg/dL	0.52(0.45–0.56)	0.5 (0.45–0.64)	0.59 (0.42–0.95)

Data are reported as median (interquartile range). HDL, high-density lipoprotein cholesterol; LDL, low-density lipoprotein cholesterol; sFlt-1, soluble fms-like tyrosine kinase 1; PlGF, placental growth factor; TAC, total antioxidant capacity; Hcy, homocysteine; UA, Uric Acid; IL-6, Interleukin-6; hs-CRP, high-sensitivity C-reactive protein; GDF-15, Growth differentiation factor-15; ADMA, Asymmetric dimethylarginine. (*), *p* < 0.05 Statistical significance when comparing low-risk and high-risk no PE groups; (ꭊ), *p* < 0.05 Statistical significance when comparing high-risk no PE and high-risk PE groups.

**Table 3 antioxidants-11-01409-t003:** Influence of biomarkers on MAP.

	Coeff B	Std. Err	Coeff b	t	Sig.
(Const.)	74.312	4.229		17.571	0.000
AU	2.066	0.705	0.254	2.929	0.004
IL-6	1.114	0.365	0.265	3.052	0.003

R^2^ = 0.175, Adjusted R^2^ = 0.161. Depend. Variable: MAP, Mean arterial pressure. UA, uric acid; Hcy, homocysteine; IL-6, interleukin-6.

## Data Availability

The data presented in this study are available in the article.

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
