# Peer review of "Factors Involved in Endothelial Dysfunction Related to Angiogenic Disbalance and Oxidative Stress, in Women at High Risk of Term Pre-Eclampsia"

_antioxidants, 2022, doi:10.3390/antiox11071409_

Round 1

Reviewer 1 Report

The data reported in this study regarding the association between endothelial dysfunction and: 1) oxidative stress 2) inflammation and 3) antiangiogenic factors in pregnancy at high risk of preeclampsia, have already been widely reported in the literature.

 The authors say they chose a sample (n = 68) made up of women at high risk of PE and that none of them developed the disease. Unfortunately this must be taken for granted as the authors omit important data In fact, they do not report the week in which the birth took place, they do not report the values ​​of protenuria, nor the weight of the fetus, all factors necessary to confirm that there was no preeclampsia. Furthermore, the authors suggest that the association between biomarkers and endothelial dysfunction observed in pregnant women at high risk for PE probably already exists in the same women before pregnancy. This conclusion has no scientific support, as no studies are done before or after pregnancy.

Author Response

We thank the reviewers for their interest in improving the work presented.

All changes in the manuscript are in bold.

As the reviewers suggested,

-We have shortened the introduction and ordered the discussion in order not to be repetitive.

-We have differentiated the sections of results from the discussion. We also present the most relevant data in figures 1 and 2.

-We add the next paragraph  at the end of introduction: The aim of this work is to study the existence or not of alterations in the biomarkers of oxidative stress (Ac urico, homocysteine ​​and total antioxidant capacity, TAC) inflammation (IL-6, hs-CRP, and GDP-15) and endothelial function (ADMA) in plasma of pregnant women diagnosed as high risk of late PE

- As suggested by the reviewers we have re-examined the database and clinical history of the patients, and we found, that 7 high-risk patients developed preeclampsia at the time of delivery, since they showed elevated blood pressures and 6 of them had proteinuria +2 dipstick, 1 with elevated liver enzymes. We have therefore reanalyzed the data by dividing the group at high risk of preeclampsia into 2 subgroups: pregnant women diagnosed at high risk of late preeclampsia who developed the disease (n=7) and those at high risk of preeclampsia who did not develop the disease (n=61).

-In Results section it now says: 57 uncomplicated pregnant women were included as a control group (Low risk no PE) and 68 women were included in the high-risk group of term PE. 61 pregnant at high risk did not develop PE (High risk no PE) and 7 pregnant women did (High risk PE).

 -In table 1 describing the demographic variables and clinical characteristics of the participants, we added the data concerning the week of gestation at delivery, weight of the newborn and type of delivery.

-In table 2, showing the analytical parameters of the patients, we have added information on proteinuria, platelets and serum creatinine determined between 36-38 weeks of gestation, provided in their medical records. Only in 7 patients diagnosed with preeclampsia a measurement of proteinuria was obtained at the time of diagnosis prior to delivery.

-In the conclusion (abstract), we specify that women at high risk of preeclampsia may represent pregnancies with pre-existing cardiovascular risk factors that are unmasked by gestational overload. These results suggest the need and opportunity for further prospective and retrospective studies in women diagnosed with a high risk of preeclampsia, regardless of whether they develop it or not, in order to identify possible cardiovascular risk factors in these women.

-In the last paragraph, the authors conclude: The identification of circulating cardiovascular biomarkers of relevance for myocardial and coronary artery function as UA, Hcy, IL-6 and ADMA may therefore be of additional value to determine which women are at greatest risk, to better understand the pathophysiological mechanism of CVD later in life in young women, and to establish measures to prevent CVD in this population.

We cannot know for certain the cause of IVF, it may be that age is a factor, which is consistent with the increased risk of PE with age.

Reviewer 2 Report

Content suggestions:

  1. Can the authors add the results of creatinine / urea and incidence of oedemas in the studied patients ? They revealed only the values of blood pressure, but these two mentioned clinical features are the signs of preeclampsia, as well.

  2. Do the authors have any data about the platelet count and basic coagulation parameters in correlation with the prothrombotic risk ?

  3. Based on the results of the study, what would the authors recommend for the management of the patients with preeclampsia in the practice ?

The manuscript can be published after minor revision (following the response to the questions of the reviewers).

Author Response

(The authors gave the same response as above.)

Reviewer 3 Report

The purpose of Santoyo et al. was to study factors and biomarkers involved in endothelial dysfunction related to oxidative stress, angiogenic disbalance and inflammation, in women at high risk of term pre-eclampsia who do not develop the disease. I have only few questions or/and comments:

1. The Introduction section is too long. Part of the information can be shortened or moved to the discussion section.

2. However, there is no reference to the parameters assessed at work.

3. The aim of the work, besides the general one, should contain more details, especially what parameters will be studied.

4. In the material and methods section, it seems that such a detailed presentation of the methods is not necessary. However, it is worth adding what was the sensitivity of the tests and the intra and inter assay.

5. Is it possible to provide more information about the sudied women. Do the authors have information on, for example, the lipid profile performed earlier in these patients? What was the cause of IVF? Could this have increased the risk of PE?

6. Discussing the results is also too long. The information contained in the tables is then repeated throughout the text.

7. Perhaps this is an old-fashioned approach, but a separate presentation of the results and discussion would greatly facilitate the reception of this work. Also, a clear indication of whose results they are, the authors of the work or other researchers would also help.

8. Perhaps it is worth presenting these results in a graphic form to make it more legible? Or break the results into sub-chapters and the text becomes clearer.

9. It is also worth presenting more detailed conclusions, taking into account the role of individual parameters in the development of PE.

Author Response

(The authors gave the same response as above.)
